# Unmutated IGHV1-69 CLL Clone Displays a Distinct Gene Expression Profile by a Comparative qRT-PCR Assay

**DOI:** 10.3390/biomedicines10030604

**Published:** 2022-03-04

**Authors:** Selena Mimmi, Domenico Maisano, Vincenzo Dattilo, Massimo Gentile, Federico Chiurazzi, Alessandro D’Ambrosio, Annamaria Zimbo, Nancy Nisticò, Annamaria Aloisio, Eleonora Vecchio, Giuseppe Fiume, Enrico Iaccino, Ileana Quinto

**Affiliations:** 1Laboratory of Immunology, Department of Experimental and Clinical Medicine, Magna Graecia University of Catanzaro, 88100 Catanzaro, Italy; maisano@unicz.it (D.M.); am.zimbo@unicz.it (A.Z.); nancynistico@unicz.it (N.N.); aloisio@unicz.it (A.A.); eleonoravecchio@unicz.it (E.V.); fiume@unicz.it (G.F.); iaccino@unicz.it (E.I.); quinto@unicz.it (I.Q.); 2Laboratory Genetics Unit, IRCCS Centro San Giovanni di Dio Fatebenefratelli, 25125 Brescia, Italy; 3Hematology Unit, Department of Onco-Hematology, A.O of Cosenza, 87100 Cosenza, Italy; massimogentile@virgilio.it; 4Hematological Clinic, Department of Clinical Medicine, University of Naples Federico II, 80131 Naples, Italy; fchiuraz@unina.it (F.C.); alexander.dambrosio@gmail.com (A.D.)

**Keywords:** B-cells, chronic lymphocytic leukemia, gene expression analysis, CLL heterogeneity, B-lymphoproliferative disorders, phage display, cell sorting

## Abstract

Chronic Lymphocytic Leukemia (CLL) is a heterogeneous disease characterized by variable clinical courses among different patients. This notion was supported by the possible coexistence of two or more independent CLL clones within the same patients, identified by the characterization of the B cell receptor immunoglobulin (BcR IG) idiotypic sequence. By using the antigen-binding site of the BcR IG as bait, the identification and isolation of aggressive and drug-resistance leukemic B-cell clones could allow a deeper biological and molecular investigation. Indeed, by the screening of phage display libraries, we previously selected a peptide binder of the idiotypic region of CLL BCR IGs expressing the unmutated rearrangement IGHV1-69 and used it as a probe to perform a peptide-based cell sorting by flow cytometry in peripheral blood samples from patients with CLL. Since the IGHV1-69 clones persisted during the follow-up time in both patients, we explored the possibility of these clones having acquired an evolutive advantage compared to the other coexisting clones in terms of a higher expression of genes involved in the survival and apoptosis escape processes. To this end, we studied the expression patterns of a panel of genes involved in apoptosis regulation and in NF-kB-dependent pro-survival signals by comparative qRT-PCR assays. According to the results, IGHV1-69 clones showed a higher expression of pro-survival and anti-apoptotic genes as compared to the other CLL clones with different immunogenetic characteristics. Moreover, these IGHV1-69 clones did not carry any characteristic genetic lesions, indicating the relevance of our approach in performing a comprehensive molecular characterization of single tumor clones, as well as for designing new personalized therapeutic approaches for the most aggressive and persistent tumor clones.

## 1. Introduction

Chronic Lymphocytic Leukemia (CLL) is a B-lymphoproliferative disorder, characterized by the proliferation and accumulation of CD19/CD5-positive B cells in the peripheral blood, lymphatic organs, and bone marrow of patients [1]. Due to the heterogeneity in clinical manifestations and clinical course, some patients remain asymptomatic for prolonged periods (so-called indolent CLL) with no need for pharmacological treatment of any kind, while other patients experience a very aggressive form of the disease and need early anti-tumor treatment, being often resistant (so-called progressive CLL) [2].

Within the panel of biomarkers supporting the diagnostic and prognostic evaluation of the disease, the assessment of the mutational status of the clonotypic immunoglobulin heavy variable (IGHV) genes has recently become noteworthy [3]. Indeed, the somatic hypermutation (SHM) status of the clonotypic B cell receptor immunoglobulin (BcR IG) can be assessed by sequencing and compared against the respective germline IGHV gene sequence [4]. The <98% deviation from the germinal sequence leads to the identification of cases with mutated IGHV genes (mutated CLL, M-CLL); vice versa, we refer to cases with an IGHV germline identity of ≥98% as unmutated CLL (U-CLL) [5]. Several pieces of evidence have shown that M-CLL patients experience a more favorable disease prognosis than U-CLL patients [6]. Thus, the accurate assessment of the SHM status is indicative of prognosis and used as a standard test before treatment, together with other mandatory tests such as bone marrow aspirate and clinical serum biochemistry [7,8].

It is commonly recognized that CLL is a BcR IG-dependent neoplasm, with antigen stimulation being a major protagonist in the pathogenesis of the disease [9]. This hypothesis was supported by the identification of a restricted repertoire of IGHV expressed by the leukemic cells [10]. Indeed, IGHV1, IGHV3, and IGHV4 gene subgroups showed to dominate the immunogenetic repertoire of CLL [11], resulting in the formation of quasi-identical BcR IGs and defining the concept of BcR IG stereotypy [6,12]. Stereotyped BcR IG were initially identified in 2004 by Messmer et al., through the presence of common, recurrent, and highly conserved motifs within the variable heavy complementarity determining region 3 (VH CDR3) [13]. To date, a continuously increasing amount of evidence supports the notion that stereotyped CLL subsets are considered highly relevant from a prognostic and predictive point of view, since their presence can be indicative for the clinical course of the disease as well as its response to therapy [12].

Moreover, it is well documented that some CLL BcR IGs may cross-link in correspondence to an intrinsic motif and trigger a type of cell-autonomous, intracellular signaling cascade, inducing Ca^2+^ influx and Nf-κB targeting without the involvement of any exogenous antigens [14].

Tumor heterogeneity plays a crucial role in disease progression [15]. In this context, the possible coexistence of two or more CLL clones in a small fraction of CLL patients may complicate things even more, given that these patients were characterized by variable levels of survival and responsiveness to treatment [16]. In this context, the BcR IG represents an optimal biomarker for assessing cases with multiple B cell clones, since it is considered to be a unique molecular signature for each individual B cell clone [17]. Moreover, the phage display technology for the identification of peptide ligands for a wide range of receptors [18] could also be used to study BcR IG idiotypes in order to monitor CLL tumor cells [16,19]. Indeed, the in vitro application of the BcR IG-specific peptide ligands allowed the application of peptide-based cell sorting and isolation of CLL cells from multiple clones [20].

Following this approach, we studied in-depth two CLL patients (named CLL1 and CLL5) over a period of 2 years, demonstrating the coexistence of several independent leukemic cell clones expressing different BcR IGs sequences. Of interest, the most persistent CLL B cell clone in both patients was characterized by the expression of an unmutated IGHV1-69 gene rearrangement [16]. The other coexisting clones expressed different BcR IGs rearrangements, either mutated or unmutated, but none of them persisted during the whole follow-up period, and/or were found at high frequencies and, thus, were not further analyzed (full BcR IG sequences are available at GenBank under accession numbers MT334403 to MT334414) [16]. Then, we focused on these IGHV-169 clones and we applied phage display-guided screening. This process led to the isolation of a specific peptide binder (named p1) for IGHV1-69 BcR IGs, which was able to specifically sort the single CLL subpopulation carrying that rearrangement [20]. Comparative analysis revealed significant variations in terms of CD5 expression levels among the sorted IGHV1-69 clones and the unsorted clones (carrying a different BcR IG gene rearrangement), laying the foundation for a deeper molecular analysis [20]. Hence, we studied the expression profile of genes belonging to the NF-κB and apoptosis pathways in CLL IGHV1-69 clones versus other clones in order to identify and characterize the mechanisms that could underlie the predominance and persistence of IGHV1-69 clones. Our data showed the upregulation of anti-apoptotic genes and the down regulation of pro-apoptotic genes in CLL IGHV1-69 clones versus other clones, inducing the predominance and persistence of this clone.

## 2. Materials and Methods

### 2.1. Patients and Samples Collection

For all the experiments performed, we used isolated, frozen CD19/CD5-positive CLL cells from two CLL patients (CLL1 and CLL5), at diagnosis (samples CLL1a and CLL5a) and at the end of follow-up (samples CLL1c and CLL5c) [16]. Through peptide-based sorting, we obtained two populations, the p1-positive cells (corresponding to the unmutated IGHV1-69 clone) and the p1-negative cells (corresponding to the other clones, both mutated and unmutated) [20].

### 2.2. Validation of p1 Binding on CLL Cells

Total CLL cell population from patients (10^6^ cells/mL) were incubated with PE-conjugated anti-CD5 antibody (Miltenyi Biotec, Bergisch Gladbach, Germany) and FITC-conjugated p1 peptide (10 μg/mL) and DAPI on ice for 20 min in the dark. After washing with Perm/Wash^TM^ Buffer solution (Becton Dickinson Italia S.p.A, Milan, Italy), cells were incubated with DAPI stain solution (Thermo Fisher, Waltham, MA, USA) at room temperature for 5 min in the dark. After extensive washing, cells were mounted under a cover slip, and visualized by confocal microscopy. Pictures were captured with a Leica TCS SP2 confocal microscope with a HCX PL APO 63.0×/1.40 oil UV objective (NA1.40) in glycerol and acquired with Leica Confocal Software Version 2.61 (Wetzlar, Germany). Image manipulation was performed with Adobe Photoshop CC 2015 software (San Jose, CA, USA)

### 2.3. NF-κB and Apoptosis Related Genes

Total mRNA was extracted from the CLL cells from all samples at both timepoints (CLL1a and CLL1c, and total CLL5a and CLL5c) for both the IGHV1-69 and the other clones using TRIzol^TM^ RNA Isolation Reagents (Invitrogen, Waltham, MA, USA) and were quantified with the use of a spectrophotometer. Five hundred μg of total mRNA were reverse transcribed into cDNA using the 5X iScript^TM^ RT Superscript (BioRad, Hercules, CA, USA). A 96-well RT^2^ Profiler PCR Array NF-κB human pathway and Apoptosis were used (Qiagen, Hilden, Germany). The reaction conditions were as follows: initial denaturation step at 95 °C for 10 min, 40 cycles of denaturation at 95 °C for 15 s, annealing at 57 °C for 30 s, and elongation at 72 °C for 30 s. All reactions were performed in triplicate employing the CFX96 Touch Deep Well Real-Time PCR System (BioRad, Hercules, CA, USA). Results were normalized with the β2 microglobulin gene (*B2M*). Expression levels were represented as ΔCt (Ct B2M gene—Ct target gene) values in the heatmap figure, and as log10 ΔCt values ± SD or log10 fold expression (2-ΔΔCt method) ± SD of triplicate assessments for the specific gene histograms. For Real-Time PCR analysis, statistical significance was evaluated using ordinary one-way analysis of variance (ANOVA), followed by Bonferroni’s test for multiple comparisons. Bars show mean values ± 95% confidence intervals based on three biological replications.

## 3. Results

We analyzed the total CLL cell population from: (i) patient CLL1 at diagnosis (CLL1a: collected at month 1—Binet stage A—IGHV1-69 clone accounting for 60% of the total repertoire) and at a later timepoint (CLL1c: collected at month 8—Binet stage C—IGHV1-69 clone representing 80% of the total repertoire), (ii) patient CLL5 at diagnosis (CLL5a: collected at month 1—Binet stage A—IGHV1-69 clone representing 75% of the total repertoire), and at a later timepoint (CLL5c: collected at month 24—Binet stage A—IGHV1-69 clone accounting for 35% of the repertoire) [20]. Both patients carried a biallelic *del*(*13q14*) involving the locus *D13S319*. No other lesions were identified by cytogenetic analysis. Clinical and biological features of analyzed samples were reported in Table 1.

Starting from the well characterized peptide p1 identified by phage display and its ability to bind and target the CLL cells carrying the BcR IG rearrangement IGHV1-69, we performed a confocal microscopy assay using the total CLL cell population from patients as a sample. Figure 1 is a representative image of CLL1c patient, demonstrating the capability of p1 peptide to bind CLL cells. Once validated, the specificity of p1 to bind tumor B cells, discriminating a clone versus total population, we performed a peptide-based cell sorting in two individual CLL samples (named CLL1c and CLL5a) using the p1 peptide (the specific peptide binder for IGHV1-69 clones) as a probe, in order to study the single CLL clones. This process resulted in a pool of p1-positive cells (IGHV1-69 clones) and a pool of p1-negative cells (non-IGHV1-69 clones). The applied workflow is illustrated in Figure 2. We chose to analyze samples CLL1c and CLL5a because they had the highest percentage of IGHV1-69-expressing cells, as reported in Table 1. Furthermore, we decided to pool together the p1-positive cell derived from samples CLL1c and CLL5a due to the small total number of cells. So, we performed comparative qRT-PCR assays on a panel of about 100 genes involved in apoptosis regulation and NF-kB pathway.

As expected, qRT-PCR results showed a constitutive activation of the NF-κB pathway as well as high expression levels for *BCL2*, *BIRC3*, and *NFKB1*, which represent well-documented de-regulated genes in the pathogenesis CLL through their involvement in the apoptosis escape process. Comparative analysis did not reveal significant differences regarding the expression levels of genes involved in pro-survival signaling between the longitudinal samples of patient CLL5, perhaps due to the indolent form of the disease observed in this case. In contrast, an increase in the expression levels of pro-survival genes was detected in patient CLL1 between the two longitudinal samples, perhaps reflecting disease progression from Binet stage A at diagnosis to Binet stage C at follow-up, indicating a more resistant and aggressive behavior of tumor cells (Figure 3). Among others, the genes showing the highest levels of increase in their expression were *ABL1*, *CASP5*, *IGF1R*, *CCL2*, *FOS*, *CSF1*, *CXCL2*, *CXCL8*, *IL10*, *IL1A*, and *IL1B* (Figure 3).

Then, we focused on gene expression levels within the IGHV1-69 clones with respect to the other CLL clones, in order to identify possible variation not highlighted by the bulk analysis.

As shown in Figure 4A, the IGHV1-69 clones showed a significant increase in the expression of pro-survival and anti-apoptotic genes compared to the non-IGHV1-69 clones, despite coexisting in the same patient. Among others, the most differentially expressed genes in the IGHV1-69 clones compared to the non-IGHV1-69 clones were *IGFR1*, *CCL2*, *IL-1α*, *CSF1*, *IL-1β*, *CXCL2*, *IL-10*, *CASP5*, *NFKB1*, *BCL2L1*, *CXCL8*, *ABL1*, and *FOS*.

To validate the advantages of single clone profile analysis compared to bulk CLL cell analysis, we compared the gene expression profiles of IGHV1-69 clones in respect to the total cell populations from both patients (samples CLL1c and CLL5a were considered for this analysis, due to the fact that they displayed a higher and similar relative frequency of IGHV1-69 cell clones) (Figure 4B,C). Against the total CLL1c population, we observed a highly statistically significant increase in the expression of *BCL2*, *BCL2L1*, *CASP5*, *IGF1R*, *NFKB1*, *CCL2*, *CSF1*, *CXCL2*, *IL10*, and *IL1A* genes in IGHV1-69 clones (Figure 4B). The same genes were also more highly expressed in sample CLL1c with respect to the sample CLL1a, but their increase was even more significant in the sorted IGHV1-69 clones.

The comparison of the gene expression profile of IGHV1-69 clones against the total CLL5a population revealed a significant increase in the expression levels of anti-apoptotic genes and a decrease of pro-apoptotic genes, with the highest variation observed for *CASP5*, *ILL1A*, *IL1B*, *IL10*, *CSF1*, and *IGF1R* (Figure 4C). For a better evaluation, we included the fold expression differences of analyzed genes in the Appendix A.

The upregulation of anti-apoptotic genes together with the downregulation of pro-apoptotic ones in IGHV1-69 clones could likely contribute to their predominance and persistence among the heterogeneous CLL cell population.

## 4. Discussion

CLL is characterized by variable clinical manifestations ranging from indolent to aggressive and often refractory to therapy [20]. This variable clinical course is likely dependent on the heterogeneity of CLL populations, characterized by the coexistence of two or more tumor clones and the possible selection of an aggressive drug-resistance tumor cell clone [21].

Single tumor clones do not show the same growth dynamics, influencing the clinical status of the patients. Indeed, in some patients the most advantageous clone stabilizes over time, whereas in other patients, one or more clones show an exponential-like growth pattern, clinically defined by the metric “short lymphocyte doubling time”, which is associated with a poor prognosis [22]. Moreover, patients with exponential cell growth show a higher number of clonal and subclonal driver gene mutations that correlate with aggressive clinical phenotypes, such as *TP53* or *NOTCH1* mutations which drive to Richter transformation [23]. Another biomarker related to “short lymphocyte doubling time” is the unmutated status of the clonotypic IGHV gene of the BcR IG, which is also associated with aggressive disease [7,24]. The present study further supported this information despite showing a clear predominance of an IGHV1-69 clone at diagnosis (relative frequency: 75%), CLL5 was characterized by a decrease in the relative size of the IGHV1-69 clone and remained stable during the whole follow-up period, without any need for therapy. In contrast, CLL1 showed an exponential growth of the IGHV1-69 clone (from 60% to 80%) during the observation period, which was reflected at the clinical level through a shift from Binet stage A to C and requiring chemotherapy.

The heterogeneity of CLL disease also may depend by the presence of subclonal mutations, which could determine different responses to treatment among patients [25]. Furthermore, while clonal mutations represent early events responsible for the neoplastic transformation, subclonal mutations could be related mostly to disease progression [26]. In this regard, Landau et al. (2013) identified three driver mutations as clonal—*MYD88*, trisomy 12, and hemizygous *del*(*13q*)—appearing in the CLL-originating cell, while other mutations appeared later during the neoplastic development, such as mutations in the *ATM*, *TP53*, *RAS*, and *SF3B1* genes, and were usually associated to poor prognosis [27]. Enlarging the set of probed genes, Landau et al. (2015) identified other driver gene mutations, such as those affecting: (i) the *FUBP1*, *XPO4*, *EWSR1*, and *NXF1* genes involved in RNA processing and export, (ii) the *CHEK2*, *BRCC3*, *ELF4*, and *DYRK1A* genes involved in DNA damage control, (iii) the *ASXL1*, *HIST1H1B*, *BAZ2B*, and *IKZF3* genes involved in chromatin modification and, (iv) the *TRAF2*, *TRAF3*, and *CARD11* genes involved in B-cell related pathways [28].

Next to the clonal and subclonal mutations, the mutational status of BcR IG is another key prognostic biomarker in CLL increasing the heterogeneity of the disease; in this regard, it is well documented that CLL expressing unmutated BcR IGs are often associated with poor prognosis and fatal outcomes, with respect to their mutated counterparts [29,30]. In this scenario, it has been hypothesized that U-CLL pathogenesis may depend on the abnormal antigen stimulation of the clonotypic BcR IG, causing deregulated intracellular signaling [31] and resulting in a constitutive B-cell clone activation [32,33,34].

Combining all these data could be very important and helpful for a better patient-specific therapy, associating the IGHV mutational status (or the BcR IG rearrangement) to the genetic alterations. In this regard, most driver gene mutations were found in U-CLL, while only *del*(*13q*), *MYD88*, and *CHD2* mutations were more frequent in M-CLL [28].

Additional interactions with the tumor microenvironment, including functional crosstalk with cells of the immune response and soluble molecules, can further sustain tumor cell activation leading to their expansion [35,36]. Indeed, cells like macrophages or T cells, as well as stromal follicular dendritic cells could stimulate survival and pro-proliferative pathways in CLL cells, both by secreting chemokines or cytokines, and by the expression of specific surface molecules [37].

All these variabilities influence the choice of therapy and the patient’s response. Common first-line treatments in CLL include combinations of purine analogues (i.e., fludarabine) plus cyclophosphamide or the addition of the anti-CD20 monoclonal antibody rituximab to fludarabine plus cyclophosphamide (FCR) [38]. More recently, two novel inhibitory molecules were introduced, such as ibrutinib and idelalisib, which are two kinase inhibitors that target Bruton tyrosine kinase and phosphatidylinositol-3-kinase, respectively, and are recommended for patients carrying *TP53* aberrations [39]. Moreover, the Bcl-2 inhibitor, venetoclax, is used for the treatment of patients with *TP53* gene aberrations in relapse or patients with contraindications for kinase inhibitors or, lastly, non-responder patients to ibrutinib or idelalisib in combination with anti-CD20 antibody [40,41]. The second- and third-line treatment strategy depends on both the response to the first-line strategy and to genetic defects due to clonal and subclonal mutations enrichment after therapy [42,43].

All these data support the need for a deeper knowledge of genetic alterations in CLL in order to improve the strategy of applying personalized treatments for patients, due to the variability of clinical and biological manifestations in CLL. Based on this evidence and considering the importance of immunogenetic characteristics exemplified by BcR IG stereotypy, our intent was to associate the assemblage of particular IGHV genes with a specific gene expression profile in order to better characterize the CLL cells.

Exploiting two previously validated CLL patients with an opposite clinical course [16], we firstly performed a comparative qRT-PCR gene profile analysis between the two patients. We observed significantly higher levels of expression of anti-apoptotic genes in the patient shifting from Binet A to Binet C stage, while no statistical differences were observed in the “stable” patient.

Subsequently, since we observed the persistence of the CLL clone identified by the unmutated IGHV1-69 gene rearrangement in both patients, we searched for the deregulated genes that could be responsible for the persistence of this clone. In this regard, Kienle et al. (2005) already reported an upregulation of *FOS* and a downregulation of *BLNK*, *SYK*, *CDK4*, and *TP53* in IGHV1-69 cases compared to M-CLL [44].

To this end, we first successfully performed a peptide-based single clone sorting, exploiting the high specificity of the p1 peptide to bind the IGHV1-69 clones, and then a comparative qRT-PCR gene profiler analysis. We had already found significant differences regarding CD5 gene expression between the IGHV1-69 clone and the others, despite CD5 being a well-known biomarker in CLL [20]. So, we focused on well-known genes involved in CLL pathogenesis, such as apoptosis related genes (namely *DAPK1*, *BCL2L1*, *BCL2L2*, *XIAP*, *ABL1*, *BCL2*, *BAG3*, *BAD*, *BAX*, *AIFM1*, *HRK*), NF-kB dependent genes (*FOS*, *BIRC3*, *NFKB1*), microenvironment related genes (*CSF1*, *CD83*, *CD40*, *IGF1R*) and inflammatory genes (*IL8*, *CCL2*, *CXCL2*, *CASP5*, *IL10*, *IL1B*, *IL1A*).

The gene expression analysis of IGHV1-69 clone, compared with the non-IGHV1-69 clones, revealed the upregulation of anti-apoptotic, inflammatory, and NF-kB related genes in the former, suggesting an apoptosis escape mechanism of this peculiarly persistent clone. In particular, we noticed the significant overexpression of *BCL-2*, *BCL2L1*, *BCL2L2*, and *XIAP*.

Our data demonstrated that cells expressing clonotypic BcR IGHV1-69 rearrangement display a distinct gene expression profile differently to the non-IGHV1-69 population and are not detectable by bulk analysis. Furthermore, the overexpression of anti-apoptotic and NF-kB related genes, in addition to the downregulation of the pro-apoptotic genes, may have rendered IGHV1-69 clones more persistent, being detectable during the whole follow-up period in both patients. Moreover, the differential manifestation of the disease, being progressive in patient CLL1 and indolent in patient CLL5, despite the dominance of an IGHV1-69 clone at diagnosis in both cases, suggests the potential involvement of other mechanisms, such as the crosstalk with the tumor microenvironment.

## 5. Conclusions

Collectively, the reported pieces of evidence represent a further advance in the validation of specific idiotype peptides as tools for monitoring neoplastic progression B and molecular characterization of individual tumor clones. Certainly, translating our peptide-devoted approach to BcR IGs commonly expressed in CLL could lead to a more informative patient classification focused on analyzing the gene expression profiles and deregulated genes associated with specific BcR IG rearrangements. Moreover, this work could provide additional information for improving the clinical management and personalized treatment of CLL patients.

## Figures and Tables

**Figure 1 biomedicines-10-00604-f001:**
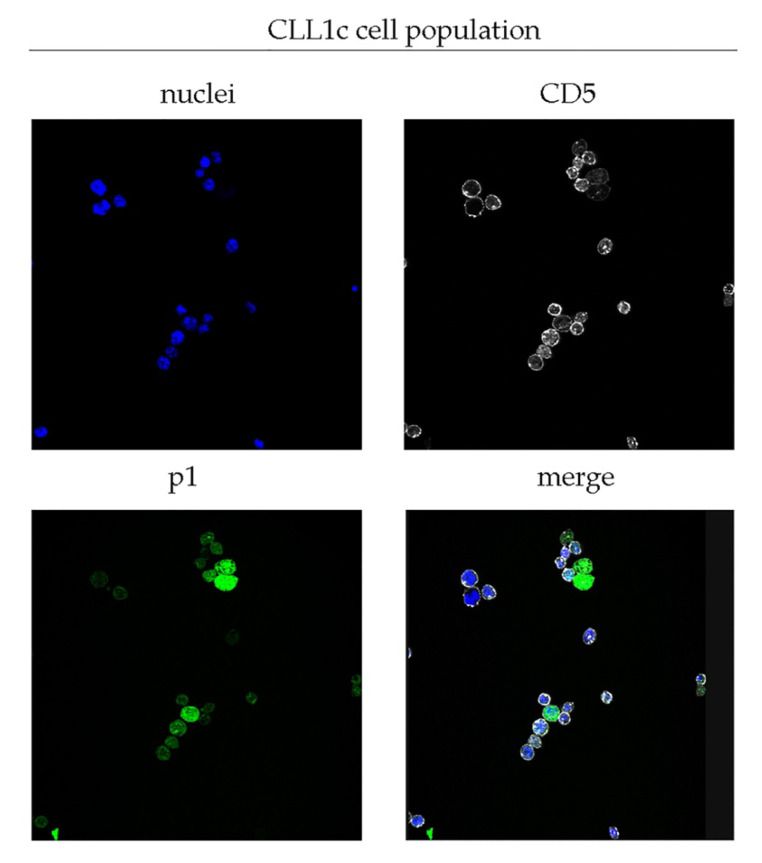
Representative confocal images of total CLL cells from CLL1c patient. Cells were labeled with DAPI to detect the nuclei (blue), anti CD5 (white) to identify the leukemic population and p1 peptide (green). The analysis was performed using a Leica TCS SP2 confocal microscope at 40× magnification.

**Figure 2 biomedicines-10-00604-f002:**
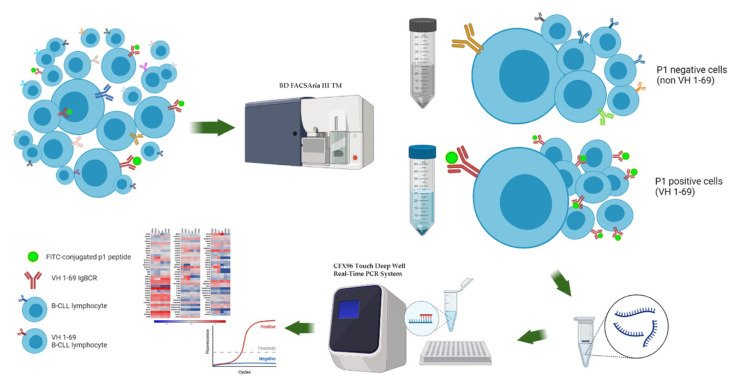
Representative workflow for the analysis of single tumor clones. The previously identified peptide p1 [16] was used as a probe to specifically target and isolate the IGHV1-69 clones among the total tumor population in the CLL1 and CLL5 patients. Two populations were obtained in each case: the p1 positive clones expressing an IGHV1-69 rearrangement, and the p1 negative clones carrying immunogenetically different IGHV gene rearrangements. Total mRNA was purified, retro-transcribed in cDNA, and analyzed by qRT PCR array for both cell populations from each patient.

**Figure 3 biomedicines-10-00604-f003:**
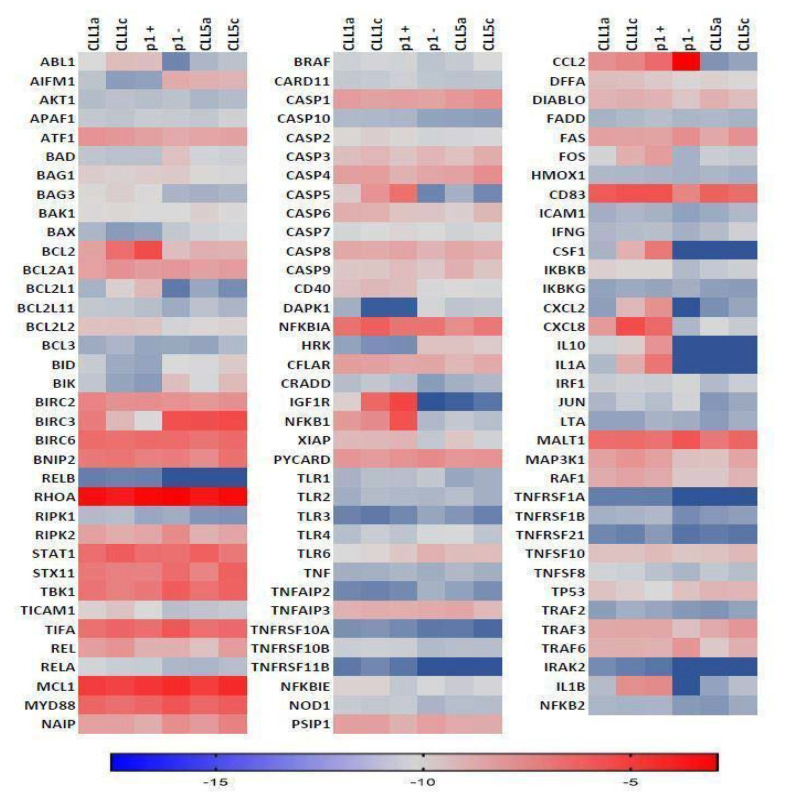
Heatmap of gene expression panel in bulk CLL1 (a, c) bulk CLL5 (a, c), p1-positive (p1+) cells, and p1-negative (p1−) cells. CLL1a: CLL1 at month 1, indolent Binet A stage; CLL1c: CLL1 at month 8, aggressive Binet C stage; CLL5a: CLL5 at month 1, indolent Binet A stage; CLL5c: CLL5 at month 24, indolent Binet A stage; p1+: p1-positive, IGHV1-69 B-CLL cells; p1−: p1-negative, non- IGHV1-69 B-CLL cells. ΔCt values of reference are reported in the bottom bar.

**Figure 4 biomedicines-10-00604-f004:**
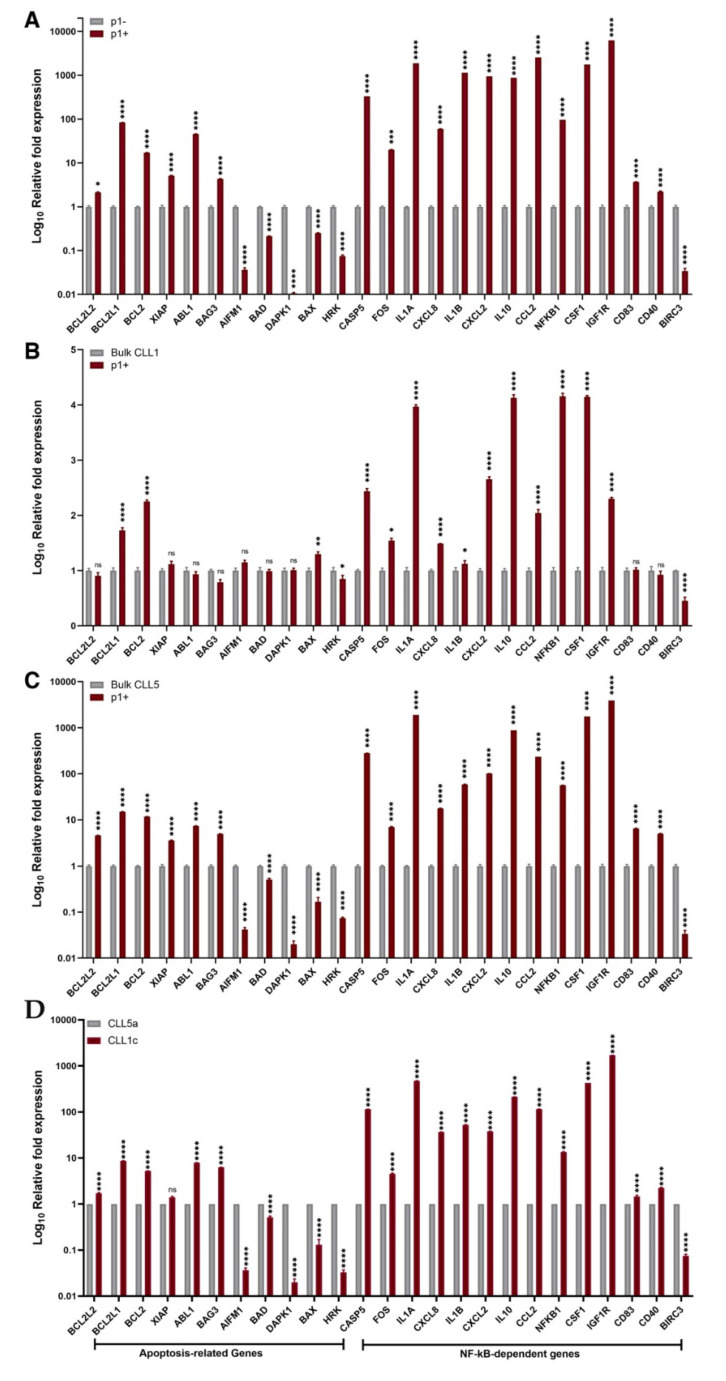
Expression levels of the most deregulated genes comparing p1-positive (p1+) cells with respect of p1-negative (p−) cells (**A**), bulk cells of CLL1c patient (**B**), and bulk cells of CLL5a patient (**C**), and comparing bulk cells of CLL1c with respect to CLL5a (**D**). Data are reported as folds ± SD of IGHV 1-69 p1+ cells versus non- IGHV 1-69 p1- cells (**A**), bulk CLL1c (**B**), and bulk CLL5a (**C**), and as folds ± SD of bulk CLL1c versus CLL5a (**D**). Statistical significance was calculated using ordinary one-way ANOVA and Bonferroni’s test of multiple comparisons. ns: not significative; * *p* ≤ 0.1; ** *p* ≤ 0.01; *** *p* ≤ 0.001; **** *p* ≤ 0.0001 of three independent experiments.

**Table 1 biomedicines-10-00604-t001:** Summary of clinical and biological features of CLL1 and CLL5 patients. Three blood samples were collected at various timepoints for each patient, but for this study we used the first and the last samples collected. CLL clones were identified by the analysis of the clonotypic BCR IG gene rearrangement through Sanger sequencing (GenBank accession numbers MT334403 to MT334414). As shown, the IGHV1-69 clone was the only one persisting in both patients during the follow-up period at various size levels in respect the total CLL population. Despite both patients carrying the same genomic aberration del(13q14), CLL1 showed disease progression through a shift from Binet stage A to C and required therapy, while the disease remained stable in patient CLL5 (Binet classification is in accordance with Cancer.net Editorial Board, October 2017).

Patient	Sample (Collection Time)	WBC (% of CD19^+^/CD5^+^)	Binet Stage	IGHV Rearrangement (Mutational Status)	(%)/Total CLL Population	Cytogenetic Alteration
CLL1 65-years old male	CLL1a (month 1)	40,410/mmc (90%)	A	V1-69 (U-CLL) V4-4 (U-CLL)	60 40	del13q14
	CLL1c (month 8)	92,970/mmc (99%)	C	V1-69 (U-CLL) V4-59*08 (U-CLL) V5-10*03 (U-CLL)	80 10 10	
CLL5 80-years old woman	CLL5a (month 1)	57,210/mmc (95%)	A	V1-69 (U-CLL) V3-7*03 (M-CLL)	75 25	del13q14
	CLL5c (month 24)	86,500/mmc (96%)	A	V1-69 (U-CLL) V3-49 (U-CLL) V4-4*02 (U-CLL) V3-7*03 (M-CLL)	35 25 25 15	

## Data Availability

Full IgBCR sequences are available at GenBank accession numbers MT334403 to MT334414 (https://www.ncbi.nlm.nih.gov/genbank/ accessed on 10 January 2022).

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
