# Peer review of "Unmutated IGHV1-69 CLL Clone Displays a Distinct Gene Expression Profile by a Comparative qRT-PCR Assay"

_biomedicines, 2022, doi:10.3390/biomedicines10030604_

Round 1
Reviewer 1 Report
line 51: "<98%% deviation" please rephrase more clearly
line 67: "highly prognostically and relevant from a prognostic" redundant phrasing
line 148: "accounting for 25% of the repertoire" Table 1 says 35%
lines 160-174: please check phrasing and language
Figure 1:
The "merge" image (lower right) has been resized. please resize to match the other 3 images
Staining of p1 peptide looks rather diffuse and non-binary. Please provide a FACS histogram marking the distinction between p1-negative and p1-positive cells
Figure 2 is not necessary. The experimental procedure is clearly explained in the methods section.
Figure 3:
The order of genes seems random, or partly alphabetic. It is thus very hard to find the genes mentioned in the Results section.
Please reorder the heatmap either by differential expression (and mark pro- and antiapoptotic genes) or by functional similarities.
In general, the analysis is interesting and contributes to the understanding of clonal evolution and selection in the heterogenous CLL disease.
The presentation however should be more organized and the expression profile of the IGHV1-69 clone should be more clearly presented.
Author Response
Dear Reviewer,
Thank you so much for your time spent on our manuscript, we have appreciated it!
Here below are the point-to-point answers to your comments:
line 51: "<98%% deviation" please rephrase more clearly
A: Thanks for the suggestion, it was a mistake. The second % was removed
line 67: "highly prognostically and relevant from a prognostic" redundant phrasing
A: Thanks for the suggestion. “prognostically and” were removed.
line 148: "accounting for 25% of the repertoire" Table 1 says 35%
A: Thanks for the suggestion, it was a mistake. The percentage was corrected.
lines 160-174: please check phrasing and language
A: Thanks for the suggestion. Checking and corrections were made.
Figure 1:The "merge" image (lower right) has been resized. please resize to match the other 3 images. Staining of p1 peptide looks rather diffuse and non-binary. Please provide a FACS histogram marking the distinction between p1-negative and p1-positive cells
A: Thanks for the suggestion. Figure 1 was modified. According to FACS data, a plot showing the identification of two distinct populations (p1 positive and p1 negative gates) is already reported in Figure 2 of the published paper Maisano et al Fronties in Oncology, 2021 (PMID: 34222027).
Figure 2 is not necessary. The experimental procedure is clearly explained in the methods section.
A: Thanks for the suggestion, and we are very sorry that you do not like it, but Figure 2 should be considered as Graphical Abstract.
Figure 3: The order of genes seems random, or partly alphabetic. It is thus very hard to find the genes mentioned in the Results section. Please reorder the heatmap either by differential expression (and mark pro- and antiapoptotic genes) or by functional similarities.
A: Thanks for the suggestion. For the heatmap, we used the same genes order of the qRT-PCR plates used, due to the use of the manufacturing software. It was a very hard work considering the number of genes analyzed. We ha reorder the genes at lines 105-106 according to the heatmap. However, statistically relevant genes discussed in the Results section were reordered according to their functions in Figure 4.
In general, the analysis is interesting and contributes to the understanding of clonal evolution and selection in the heterogenous CLL disease.The presentation however should be more organized and the expression profile of the IGHV1-69 clone should be more clearly presented.
A: Thanks for your comment, we are very glad you have appreciated our work.
We sincerely hope to have satisfied your requests.
Reviewer 2 Report
The results presented in the manuscript are interesting and are deserved to be published after the major revision.
First, the conclusions I have made, analyzing the presented results, differ from the author’s conclusions.
(1.) I am completely agree with the author’s conclusion (lines 322-326): “Exploiting two previously validated CLL patients, who both expressed the BcR IG rearrangement of the IGHV1-69 gene [16], we performed a comparative qRT-PCR gene profile analysis and observed significantly higher levels of expression of anti-apoptotic genes during the clinical progression of the disease, as well as between cases characterized by indolent and aggressive disease.”
(2.) I am convinced that, in the case of the patient CLL1 with the aggressive disease,:
(2.1.) (lines 340-343) “The gene expression analysis of IGHV1-69 clone, compared with the non-IGHV1-69 clones, revealed the up-regulation of anti-apoptotic, inflammatory and NF-kB related genes in the former, suggesting an apoptosis escape mechanism of this peculiarly persistent clone.”
(2.2.) (lines 225-228) “Against the total CLL1c population, we observed a highly statistically significant increase in the expression of BCL2, BCL2L1, CASP5, IGF1R, NFKB1, CCL2, CSF1, CXCL2, IL10, and IL1A genes in IGHV1-69 clones (Figure 4, panel B).” Moreover, this “highly statistically significant increase” is 2 – 4 fold (according to the OY axis values).
(3.) In the case of the patient CLL5 with the indolent disease, the presented data - Expression levels of the most deregulated genes (Figure 4), are not convincing, because:
(3.1.) The p1-positive (p1+) cells – these are the pooled cells from both CLL1c (with aggressive disease at the advanced stage C) and CLL5a (with indolent disease at the stage A), in about equal proportion.
However, the authors have “observed significantly higher levels of expression of anti-apoptotic genes during the clinical progression of the disease, as well as between cases characterized by indolent and aggressive disease” (lines 322-326).
(3.2.) In Figure 4, an additional plot must be included demonstrating the expression levels of the most deregulated genes comparing bulk cells of the CLL1c patient and bulk cells of the CLL5a patient.
Then, the fold increase in bulk cells of the CLL1c relative to bulk cells of the CLL5a must be estimated and compared to the fold increase in the pooled p1-positive (p1+) cells relative to bulk cells of the CLL5a (Figure 4C).
Author Response
Dear Reviewer,
Thank you so much for your time spent on our manuscript, we have appreciated it!
Here below are the point-to-point answers to your comments:
The results presented in the manuscript are interesting and are deserved to be published after the major revision.
First, the conclusions I have made, analyzing the presented results, differ from the author’s conclusions.
(1.) I am completely agree with the author’s conclusion (lines 322-326): “Exploiting two previously validated CLL patients, who both expressed the BcR IG rearrangement of the IGHV1-69 gene [16], we performed a comparative qRT-PCR gene profile analysis and observed significantly higher levels of expression of anti-apoptotic genes during the clinical progression of the disease, as well as between cases characterized by indolent and aggressive disease.”
A: Thanks for your comment, we are very glad you have appreciated our work.
(2.) I am convinced that, in the case of the patient CLL1 with the aggressive disease,:
(2.1.) (lines 340-343) “The gene expression analysis of IGHV1-69 clone, compared with the non-IGHV1-69 clones, revealed the up-regulation of anti-apoptotic, inflammatory and NF-kB related genes in the former, suggesting an apoptosis escape mechanism of this peculiarly persistent clone.”
(2.2.) (lines 225-228) “Against the total CLL1c population, we observed a highly statistically significant increase in the expression of BCL2, BCL2L1, CASP5, IGF1R, NFKB1, CCL2, CSF1, CXCL2, IL10, and IL1A genes in IGHV1-69 clones (Figure 4, panel B).” Moreover, this “highly statistically significant increase” is 2 – 4 fold (according to the OY axis values).
A: Thanks for your comment, we are very glad you have appreciated our work.
(3.) In the case of the patient CLL5 with the indolent disease, the presented data - Expression levels of the most deregulated genes (Figure 4), are not convincing, because:
(3.1.) The p1-positive (p1+) cells – these are the pooled cells from both CLL1c (with aggressive disease at the advanced stage C) and CLL5a (with indolent disease at the stage A), in about equal proportion.
A: As we highlighted in the text “we decided to pool together the p1-positive cell derived from samples CLL1c and CLL5a due to the small total number of cells”, just based on the IGHV rearrangement.
However, the authors have “observed significantly higher levels of expression of anti-apoptotic genes during the clinical progression of the disease, as well as between cases characterized by indolent and aggressive disease” (lines 322-326).
A: Thank you for the suggestion. We agree with you; this sentence has been corrected removing “as well as between cases characterized by indolent and aggressive disease”.
(3.2.) In Figure 4, an additional plot must be included demonstrating the expression levels of the most deregulated genes comparing bulk cells of the CLL1c patient and bulk cells of the CLL5a patient.
A: Thank you for the suggestion. The panel was added as Figure4D together with the relative legend.
Then, the fold increase in bulk cells of the CLL1c relative to bulk cells of the CLL5a must be estimated and compared to the fold increase in the pooled p1-positive (p1+) cells relative to bulk cells of the CLL5a (Figure 4C).
A: Thank you for the suggestion. A table summarizing the folds was added as supplementary data (Supplementary Table 1)
We sincerely hope to have satisfied your requests.
Round 2
Reviewer 1 Report
The issues have been addressed with minimal effort.
The presentation is still very confusing and hard to follow, which will diminish the possibility of being cited by other researchers.
However, I do not see any issues regarding scientific soundness.
Author Response
Thank you for your comments.
We have modified some parts of the text, trying to address the major issue of the work, that is the heterogeneity of the disease, that requires a deeper investigation for better patient-specific treatment.
Reviewer 2 Report
I am satisfied with the additions that the authors have made to the manuscript.
Some minor requests are left.
- There are no references to the Supplementary data in the article body text.
- In Supplements, first two Figures are the duplicates: the first one is the duplicate of the Figure 4D; the second one is the duplicate of the next, the third one in Supplements (also, the legend is to the Figure 4C).
3. My principal request (lines 164-166): it is written, “In addition, Figure 1 highlighted how p1 peptide (green) did not bind all total CD5 positive cell population, supporting the specificity of p1 peptide to bind only the IGHV1-69 clones.”
The underlined part must be removed, because the staining do demonstrate that p1 binds CD5+ cells of CLL patients, but the authors do not present data that these CD5+/p1+ cells are the only IGHV1-69 clones, nor that the number of CD5+/p1+ cells in dozens analyzed images is about equal to the frequency of the IGHV1-69 clones within the total CLL population.
Author Response
Thank you for your comments. 1. We did not understand how the duplicate figures have appeared in the Supplementary file. We have removed them. 2. At the same time, we have added the references to the supplementary data (lines 236-238: For a better evaluation, we included the fold expression differences of analyzed genes in Supplementary Table 1 3. As you suggested, we have removed the sentence
Thanks again for your precious support.